# Study Models for *Chlamydia trachomatis* Infection of the Female Reproductive Tract

**DOI:** 10.3390/microorganisms13030553

**Published:** 2025-02-28

**Authors:** Jaehyeon Kim, Milena Ślęczkowska, Beatriz Nobre, Paul Wieringa

**Affiliations:** Complex Tissue Regeneration, MERLN Institute for Technology-Inspired Regenerative Medicine, Maastricht University, 6229 ER Maastricht, The Netherlands; j.kim@maastrichtuniversity.nl (J.K.); milena.molasy@maastrichtuniversity.nl (M.Ś.); beatriz.canobre@gmail.com (B.N.)

**Keywords:** *Chlamydia trachomatis*, chlamydia infection, host–pathogen interaction, model systems to study chlamydiae and chlamydia-like infections

## Abstract

*Chlamydia trachomatis* (Ct) is a leading cause of sexually transmitted infections globally, often resulting in inflammatory disorders, ectopic pregnancies, and infertility. Studying Ct’s pathogenesis remains challenging due to its unique life cycle and host-specific interactions, which require diverse experimental models. Animal studies using mouse, guinea pig, pig, and non-human primate models provide valuable insights into immune responses, hormonal influences, and disease progression. However, they face limitations in terms of translational relevance due to physiological differences, as well as ethical concerns. Complementing these, in vitro systems, ranging from simple monolayer to advanced three-dimensional models, exhibit improved physiological relevance by replicating the human tissue architecture. This includes the detailed investigation of epithelial barrier disruptions, epithelium–stroma interactions, and immune responses at a cellular level. Nonetheless, in vitro models fall short in mimicking the intricate tissue structures found in vivo and, therefore, cannot faithfully replicate the host–pathogen interactions or infection dynamics observed in living organisms. This review presents a comprehensive overview of the in vivo and in vitro models employed over the past few decades to investigate Ct and its pathogenesis, addressing their strengths and limitations. Furthermore, we explore emerging technologies, including organ-on-chip and in silico models, as promising tools to overcome the existing challenges and refine our understanding of Ct infections.

## 1. Introduction

*Chlamydia trachomatis* (Ct) is a Gram-negative bacterium that infects mainly epithelial cells in the human reproductive tract [1]. Ct is considered a leading cause of bacterial sexually transmitted infections (STIs) globally [2]. According to the World Health Organization (WHO), an estimated 128.5 million Ct infections occurred in adults in 2020, with higher prevalence in women than men aged between 15 and 49 years [3]. The most common symptoms of genital Ct infection in females are abnormal vaginal discharge, painful micturition, discomfort in the lower abdomen, and pelvic pain; however, these infections are often asymptomatic [4,5]. If left untreated, they may lead to long-term reproductive damage, including pelvic inflammatory disease (PID) [6], ectopic pregnancies [7], and tubal infertility, frequently due to scarring and the obstruction of the fallopian tubes [8].

*Chlamydia trachomatis* is an obligate intracellular pathogen with a unique life cycle and infection mechanism and can survive and replicate only inside host cells [1]. Ct exists in two forms: the infectious but metabolically inactive elementary body (EB) and the non-infectious, actively replicating reticulate body (RB) (Figure 1) [2]. The first step of infection includes EBs attaching to the host cell surface, targeting mainly heparan sulfate proteoglycan (HSPG) receptors via chlamydial OmcB, the major outer membrane protein (MOMP), and polymorphic membrane proteins (Pmps) [1,2]. After attachment, EBs trigger endocytosis and cytoskeleton rearrangement to form a vesicle around the EB, called “inclusions”, which prevents their degradation [2]. Chlamydia uses type III secretion systems (T3SSs), which alter host cell signaling and inhabit lysosomal fusion, enabling EBs to differentiate into RBs inside inclusions [1,2]. After replication, RBs re-differentiate into EBs, which are later released from infected cells and initiate new infections in surrounding cells [2]. In some stress conditions, such as the deprivation of nutrients, antibiotic presence, or host immune responses, chlamydia can differentiate into a third form called aberrant bodies (ABs) [9]. ABs are abnormal, enlarged forms of Ct, able to evade the host defenses and persist in a dormant state until the stress conditions are alleviated. Then, Ct can transform back into the RB form and complete its infectious cycle [10].

Ct infection data based on epidemiological studies, clinical studies, and population screening programs provide valuable insights into Ct’s distribution and risk factors and the effectiveness of prevention and treatment strategies [11,12,13]. However, the existence of multiple Ct serovars and the two-stage developmental cycle, which can be modulated in response to host immune responses, make researching the underlying mechanisms of chlamydia a significant challenge [14]. For more in-depth, mechanistic investigations into Ct’s pathogenesis, researchers have used various animal and in vitro models, with varying degrees of success depending on the serovar being studied.

The different serovars of *Chlamydia trachomatis* are classified based on their characteristics, tissue specificity, and disease manifestations [15]. Serovars A–C cause ocular trachoma and can lead to blindness, especially in developing countries due to poor hygiene and limited access to healthcare [16]. Serovars D–K are associated with urogenital infections and are highly transmissible through sexual contact [15]. Among them, the most prevalent strains and most frequently isolated in clinical settings are serovars D, E, and F, with serovars D and E being the most often used serovars for research on urogenital infections [11,17]. Despite being clinically relevant, these serovars do not readily trigger infections within the current animal or in vitro research models. A successful infection is often achieved within these model systems by applying supraphysiological bacterial loads or by the coadministration of infection-facilitating compounds to promote Ct uptake by the epithelial tissue [18].

Serovars L1–L3, known as lymphogranuloma venereum (LGV), are STIs that are also frequently used within models. Unlike other forms of chlamydial infections, which target the mucosal surfaces of the reproductive tract or eyes, LGV affects mainly the lymphatic system and results in more severe, invasive infections [15]. Infections of LGV have been considered rare in developed countries; however, in recent years, they have become more frequent in Europe and North America [19]. These infections are predominantly caused by L2 variants and have been reported in men who have sex with men (MSM), with significant associations with proctocolitis in HIV-positive individuals [19,20]. Although LGV infections are less frequent in women, the highly pathogenic behavior of LGV serovars, especially L2, has led to their common use in research on the female reproductive health. While LGV may not be a representative pathogen, it ensures a positive infection within model systems and has allowed us to further our knowledge of Ct infections [20,21,22,23].

Despite the advances in our understanding of Ct, there remains a lack of animal or in vitro models that enable the full replication of human Ct infections in the reproductive tract. This highlights the need for alternative models to overcome these challenges and advance our understanding of Ct’s pathogenesis. The use of animal models, while valuable for certain aspects of research, is often limited by their general suitability. Most animals are not naturally infected with human Ct serovars, which are not endemic to non-human hosts [24]. Moreover, the immune responses of these animal models have limited relevance to human infections, which creates challenges in translation [25]. In vitro cell culture systems, on the other hand, have been widely used to study Ct infection at the cellular level. While these models provide detailed insights into host–pathogen interactions within controlled environments, they lack the complexity of the tissue components and architecture; thus, they fail to represent key aspects of the infection process, such as immune responses and fibrosis, present in vivo [26,27].

Here, we present a comprehensive overview of recent trends in Ct research over the past decade, focusing on the various in vivo and in vitro models extensively employed to deepen our understanding of chlamydial pathogenesis. Additionally, we discuss the future directions of such research that will be valuable for more in-depth investigations.

## 2. In Vivo Models

Animal models have been instrumental in advancing our understanding the mechanisms of Ct infections. Mouse models are historically the most frequently used system, despite key differences from Ct infections in humans. Alternative models that are suggested to more closely mirror the clinical context include porcine, guinea pig, and non-human primate models. Below, we discuss each of these models in terms of their suitability and utility for the further study of Ct, with a summary comparison of their (patho)physiological and anatomical differences provided in Figure 2 and Table 1.

### 2.1. Mouse Models

Mouse models have been extensively adopted in *Chlamydia* research due to their small size, ease of handling, relatively low maintenance costs, and high availability compared to other species. The use of well-characterized inbred and knockout strains has also provided invaluable insights into chlamydial pathogenesis, host immune responses, and the evaluation of vaccines [28,29].

Historically, *Chlamydia muridarum* (Cm) has been used extensively to model chlamydia in mice. Cm is a natural murine pathogen found in mouse lungs that causes pneumonitis but can be introduced intravaginally in mice and shows similar impacts to acute Ct infections in women [30,31]. The virulent nature of Cm leads to a high bacterial burden, generating significant knowledge about the immune system response and long-term impacts, such as uterine dilation, oviduct fibrosis, and tubal occlusion [32,33].

However, using human Ct within a mouse model offers a more clinically representative despite the pathogen/host species discrepancy. Unlike Cm, Ct is more susceptible to interferon-γ (IFN-γ) responses, providing insights into the role of innate immunity in combating chlamydial infections [22]. Moreover, using different Ct serovars, such as serovar D or the more invasive serovar L2, allows researchers to tailor their models to specific infection outcomes [31]. This versatility makes Ct models particularly valuable in studying different stages of infection and immune responses.

A key limitation is that Ct does not naturally infect the mouse upper genital tract, as their immune system clears the infection rapidly, preventing the bacteria’s ascension [34,35,36]. Consequently, replicating the disease progression and resulting tubal pathologies observed in humans is challenging. Thus, intravaginal inoculation with Ct is primarily used to study lower genital tract infections [37]. To address this, alternative methods have been developed that bypass the cervix to specifically target the uterus or upper genital tract. An early technique was the highly invasive intrabursal approach, wherein surgery exposes the upper genital tract and chlamydia is directly injected into the oviduct [38]. Despite allowing one to investigate acute chlamydial infections of the oviduct and tubal factor infertility using Cm, the surgical challenges and the bypassing of the immune responses in the lower reproductive tract have led the intrabursal method to be largely replaced by transcervical inoculation. This shift has significantly advanced our understanding of chlamydial pathogenesis and the immune responses in the upper genital tract [31,39]. For example, Gondek et al. (2012) used transcervical inoculation in genetically modified mice to explore the role of T cells in chlamydial clearance [35]. Similarly, Stary et al. (2015) demonstrated the potential of UV-inactivated Ct as a vaccine by showing its ability to activate memory T cells after transcervical inoculation, and Pal et al. (2017) developed a transgenic mouse model expressing human leukocyte antigen DR4 (HLA-DR4), which allowed for a better understanding of the immune responses following serovar D infection [40,41].

Despite these advancements, significant challenges remain in using mouse models to study Ct infections, particularly in understanding the full impact on the oviduct. An intrinsic limitation is that laboratory mice lack genetic diversity, which increases the reproducibility of the results but also does not capture the genetically diverse human population or the pathogenic response to Ct [42]. Another major restriction is that mouse models cannot sustain a chronic Ct infection and, therefore, cannot be used to study long-term complications of this disease, such as pelvic inflammatory disease (PID), tubal factor infertility, or ectopic pregnancies [30,43]. Thus, without capturing the breadth and extent of the human response to a Ct infection, the utility of mouse models remains limited.

### 2.2. Porcine (Pig) Models

Porcine models offer a more compelling alternative to murine models in Ct research due to their closer resemblance to humans, including their comparable size, genital anatomy, and immune system [43,44,45]. Furthermore, *Chlamydia suis* (Cs), a natural strain of pigs, closely mirrors Ct in its infection mechanisms and inflammatory responses [44,46]. For these reasons, pigs have been successfully employed in genital chlamydia research and vaccine development [47,48].

Despite these similarities, there are still physiological differences that can influence the infection outcomes. For instance, pigs lack vaginal and cervical columnar epithelial cells and have long cervical canals with mucosal folds, both of which may hinder the reflection of human infections via intravaginal inoculation (Figure 2). As a result, intrauterine or transcervical inoculations have been used to better model human disease [49,50,51]. In a recent study by De Clerq et al. (2020), the intravaginal inoculation of Ct serovar L2 in pigs resulted in only mild lesions during primary infection, with severe pathologies occurring only upon reinfection [52]. In contrast, Lorenzen et al. (2017) achieved a persistent infection using the transcervical inoculation of Ct serovar D, highlighting the importance of bypassing the cervix to simulate the human pathology more accurately [51].

Further challenges for porcine models include higher costs and logistical challenges regarding space requirements for the housing and maintenance of pigs compared to smaller animal models [30]. Additionally, the development of genetic tools, such as knockout or transgenic pigs, is technically demanding, time-consuming, and less efficient than similar procedures in murine models [53]. Another limitation is the lack of available pig-specific immunological reagents, which can impede experimental readouts and necessitates more fundamental research and development efforts before the porcine model can realize its full potential [25].

### 2.3. Guinea Pig Models

Guinea pigs naturally carry the *Chlamydia caviae* (Cc) strain, which induces guinea pig inclusion conjunctivitis (GPIC). When introduced in the genital tract, Cc induces pathologies similar to human Ct infections, including ascending female reproductive tract infections [54]. Guinea pig models uniquely enable the study of sexual transmission from intraurethrally infected males to females or conjunctivitis in newborns from intravaginally infected females, supporting the theory that human congenital Ct infections originate from the maternal genital tract [55,56]. Studies have also demonstrated that guinea pigs treated with estradiol created a more susceptible environment for cervical and vaginal Cc infections, analogous to human conditions [57,58]. Such findings have made guinea pigs a preferred model for the study of chlamydial transmission, infection, and immune responses [59].

Accordingly, the usage of guinea pig models has been extended to Ct research. De Jonge et al. (2011) reported that estradiol-treated guinea pigs infected with Ct serovars D and E developed lymphocyte and neutrophil infiltration, confirming the ascending infection towards the upper genital tract [60]. Research by Arulanandam’s group highlighted the role of phosphatidylcholine, a genital tract lipidome metabolite, in enhancing inflammation during early-stage intravaginal infection with Ct serovar D in guinea pigs [61]. They further demonstrated that guinea pigs intranasally vaccinated with rCPAF/CpG and challenged with Ct serovar D exhibited significantly reduced vaginal shedding and lower genital tissue inflammation, demonstrating the model’s effectiveness for vaccine development [62].

Despite their advantages, guinea pig models face challenges. Similarly to porcine models, there is limited availability of immunological reagents for studies [59]. Additionally, while the guinea pig vaginal epithelium is comparable to that of humans, differences in the vaginal microbiota composition and abundance may pose translational issues, as the microbiota are critical in defending against pathogens and regulating reproductive tract inflammation [63,64]. Finally, their size, genital anatomy, and hormone cycles also differ widely from those of humans, presenting barriers to direct correlations between pathologies (Figure 2) [65,66].

### 2.4. Non-Human Primate Models

Non-human primate (NHP) models have been integral in investigating human diseases due to their close anatomical and physiological similarities, including their long lift span and genetical similarities (Figure 2) [67,68]. NHP models particularly benefit research on female reproductive tract (patho-)physiologies because of the similarities to human organ structures, menstrual cycles (28–35 days), hormonal patterns, and genital microbiota [64,69,70]. Specifically, NHPs are susceptible to urogenital Ct infections, which make them valuable in studying transmission and pathogenesis in both acute and chronic cases [25].

Over the past decade, the most used NHP species in Ct research have been rhesus macaques (*Macaca mulatta*), cynomolgus macaques (*Macaca fascicularis*), pigtailed macaques (*Macaca nemestrina*), and baboons (*Papio anubis*) [71,72,73,74,75,76]. The physiological compatibility of these species with human Ct infections allows for the accurate modeling of the disease and the evaluation of the efficacy and safety of potential vaccines or antimicrobial interventions. For example, the inoculation of pigtailed macaques with serovars D and E at either the oviduct or cervix led to a stronger immune reaction in the oviduct compared to the cervix, offering insights into Ct-induced infertility [71]. Furthermore, serum collection in NHP Ct models serves to monitor the biomarkers of local immune responses such as antibodies and chemokines and therefore track infection progression [77]. This enables longitudinal analyses by comparing the sera across multiple time points during different infection and treatment paradigms [75,76].

However, the use of NHPs in Ct research presents significant challenges. Ethical concerns are paramount, as NHPs are considered as highly intelligent and social animals, requiring stringent justifications for their use and the development of improved animal welfare. Additionally, the high maintenance costs and limited availability of NHPs also restrict the scale and scope of these studies [67,78]. In the European Union, the use of NHPs is strictly regulated under Directive 2010/63/EU, leading to a remarkable reduction in research involving NHPs but increased political pressure in science, as well as concerns about relocating such studies outside Europe [79].

**Figure 2 microorganisms-13-00553-f002:**
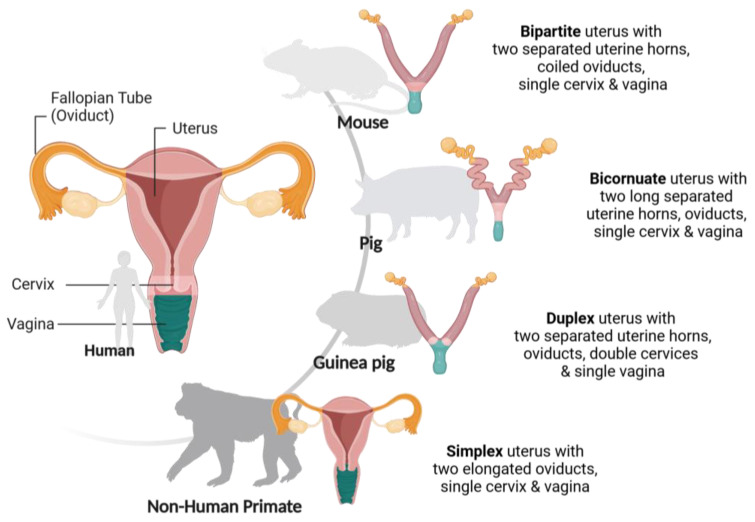
Comparative anatomy of the female reproductive tract in humans, mice, pigs, guinea pigs, and non-human primates, highlighting interspecies variations in shape and the arrangement of key reproductive structures. Inspired by [80]. Created with BioRender.com accessed on 23 December 2024.

Overall, the movement within science to have animal studies embrace the 3R principles (replacement, reduction, and refinement) has continued to motivate scientists to seek animal-free alternatives to studying Ct’s pathophysiology.

## 3. In Vitro Models

The development of in vitro models has received increased attention in the biomedical domain, often cited as a more ethical alternative to using animals for research, as well as being potentially more accurate because of the use of human cells to study the underlying mechanisms. The majority of recent developments have centered on allowing these models to better mirror the human anatomy and physiology, enabling researchers to generate findings that are more representative of the human response. Below, we draw upon general advances and concepts that apply to the current in vitro models (shown in Figure 3), including non-Ct application, and highlight the Ct-specific applications of these models, which serve to improve our understanding of this disease.

### 3.1. Cell Sources

In Ct research, in vitro models have gained significant attention as ethical alternatives to animal experiments, typically focusing on creating a functional epithelium that reflects the female reproductive tract, since the mucosal epithelium is the primary site for Ct infection [81,82]. The lower genital tract, including the ectocervix and vagina, has a stratified squamous epithelium lacking tight junctions. In contrast, the upper genital tract, comprising the endocervix, uterus, and oviduct, is lined with a monolayer of polarized columnar epithelial cells with distinct tight junctions [81,83]. These epithelia function as a physical barrier that pathogens must transverse, with dense stromal layers located underneath [84,85].

Given the severe impact of Ct infection on the upper genital tract, in vitro models of Ct infection typically aim to recreate the columnar epithelium with secretory and ciliated cells, emphasizing polarized cells with tight junctions, to study how the pathogen breaches the epithelial barrier [45,81,86]. Furthermore, the presence of diverse cell types within the epithelium model is important to create a physiologically representative epithelial barrier, as demonstrated in various in vitro systems, including intestinal epithelium studies [87]. In addition, this phenotype heterogeneity can help researchers to understand how different cell morphologies or differentiation stages may influence the infection susceptibility [81,88].

For this reason, various epithelial cell lines have been selected to create a functional epithelium for Ct research. A cell line is a population of cells that are isolated from tumors or genetically modified to be immortalized, allowing indefinite culture in the laboratory without the significant loss of its viability [89]. Commonly used cell lines in Ct study include HeLa (human cervical cancer cells), Hep-2 (a HeLa derivative), EC-1B (human endometrial adenocarcinoma cells), End1 (human endocervix epithelial cells), A2EN (human endocervix epithelial cells), and OE-E6/E7 (human fallopian tube epithelial cells) [27,90,91,92,93,94,95,96]. These cell lines are favored for their improved survival, low-cost maintenance, ease of handling, and genetic homogeneity, ensuring reproducible and consistent experimental setups over time [89,97].

While these cell lines are valuable research tools, they present certain limitations. Tumor-derived cell lines such as HeLa, Hep-2, and EC-1B are naturally immortalized cells due to the mutations that occur during tumor development [98]. These mutations frequently lead to chromosomal abnormalities, the dysregulation of apoptotic pathways, higher mutation rates, and genetic drift over passages, reducing their relevance in studying normal physiological processes [99,100]. Similarly, immortalized cell lines such as End1, A2EN, and OE-E6/E7 may exhibit different behaviors compared to native tissue observed in vivo or primary cells that are harvested from organs directly. The process of immortalization can induce the accumulation of genetic mutations that can alter cellular responses such as cytokine secretion, protein production, metabolism, and differentiation [97,101].

In particular, in vitro models of the urogenital epithelium should ideally emulate the dynamic response to female steroid hormones. In the fallopian tube, for instance, estradiol promotes epithelial differentiation, ciliogenesis, and the cilia beat frequency, while it stimulates endometrial cell proliferation and the thickening of the endometrium [102,103]. However, cell lines can exhibit the lowered expression of hormone receptors compared to native tissue [104]. This represents a major limitation because of the significant role of hormones in the tissue susceptibility to a Ct infection and the subsequent disease progression [105,106,107]. Additionally, the use of homologous cell lines often makes it difficult to replicate the complex cellular heterogeneity of native tissue. For instance, as highlighted in recent single-cell RNA sequencing studies, the fallopian tube contains multiple epithelial subtypes, which play crucial roles in tissue homeostasis and disease progression [108]. This varied cellular composition and the subsequent cell–cell interactions tend to be lost in models using cell lines. Overall, despite the contribution of cell lines to Ct research, they do not accurately capture the complex interactions between the pathogen, host cell, and hormonal milieu that is present in the upper reproductive tract [109].

In contrast, primary cells offer distinct advantages that address the limitations of cell lines, providing a diverse array of solutions to specific research questions, while cell lines are more restricted to a range of source species [100]. While human and rodent cell lines are prevalent in research, primary cells offer superior flexibility in investigating specific species or conditions, such as age, genetic status, or medical conditions, as they can be directly sourced from donors with the desired characteristics [110].

Nevertheless, their use presents significant challenges in terms of consistency, availability, and standardization. When primary cells reach a certain number of divisions, namely the Hayflick limit, the cells start to lose their capacity to replicate due to shortened telomeres [111]. Moreover, changes in the environment that create different intrinsic or extrinsic stimuli, nutrient deprivation, and oxidative or genotoxic stress can lead to replicative senescence, manifesting in phenotype changes, chromatic alternations, and metabolic dysfunction [112,113,114].

### 3.2. Two-Dimensional Models

#### 3.2.1. Traditional Mono-Cell Culture

Mono-cell culture is based on the use of a single cell type, typically grown as a monolayer attached to the bottom tissue culture platform [115]. This traditional in vitro model has significantly contributed to our knowledge of chlamydial biology over the past few decades by allowing us to examine molecular and cellular pathways and enabling rapid drug and vaccine screening [27,96].

The majority of conventional monolayer epithelial in vitro models utilize two-dimensional (2D) plastic tissue culture plates (TCPs), bringing the significant benefit of simplicity in handling, but they have inherent drawbacks in experimental setups, leading to a limited resemblance to the epithelium in vivo [116]. Mature epithelial monolayers require cellular polarization, tight junctions, and phenotype heterogeneity [117]. In particular, the Ct bacterium, as a luminal pathogen, targets the apical surface of the epithelium; thus, having a polarized surface is pivotal in building a model for Ct research [118]. However, polystyrene or polycarbonate TCPs, whose stiffness significantly exceeds that of biological tissues, can potentially prevent polarization. Epithelial cells are mechanosensitive, and their interactions with rigid substrates can not only switch the cellular polarity but also trigger epithelial–mesenchymal transition (EMT), which is a hallmark response to Ct infection [119,120,121,122]. This supraphysiological stiffness of the TCP surface may give mechanical signals to cells, leading to EMT even when chlamydia is not present. Thus, the results of TCP-based monolayer studies require careful interpretation due to substrate rigidity.

Another challenge of monolayer models is the lack of complexity that they can sustain compared to native tissues. This complexity is typically maintained via epithelial differentiation, a process orchestrated by its microenvironment, requiring the involvement of multiple signaling factors [123]. However, epithelial cells in monolayer cultures tend to lose their capacity to differentiate [124]. In particular, primary cells on monolayer cultures undergo dedifferentiation processes, which makes it difficult to preserve their morphological or functional properties [125].

Monolayer models also bring difficulties in recreating the localized initiation and progression of the disease along the urogenital tract. In vivo, Ct particles are introduced distally and progress upwards, a dynamic that traditional monolayer systems cannot adequately model. This limitation is particularly problematic when investigating the factors driving Ct’s spread from the initial infection site to the upper reproductive tract and its cell-specific targeting, as monolayer cultures, by nature, fail to capture the spatial and temporal aspects of the disease’s progression [126].

#### 3.2.2. Membrane Inserts, Including Corning Transwell^®^ and the Greiner ThinCert™

A cell culture membrane insert consists of a porous membrane that separates the culture environment into apical (upper) and basolateral (lower) compartments, enabling diverse experimental setups in one platform, such as the application of two distinct media, permeability assays, or the co-culture of various cell types [127,128,129,130].

Membrane insert systems are particularly advantageous for epithelial models as they can maintain the cellular polarity by supporting advanced techniques like air–liquid interfacing (ALI) [128,131]. ALI, which exposes the apical compartment to the air while maintaining the basolateral compartment intact with the medium, has been shown to drive epithelial cell differentiation and pseudo-stratification [81]. By establishing a concentration gradient through the basolateral diffusion of small molecules, including growth factors and gases, ALI fosters the differentiation of heterogeneous epithelial cell populations, such as mucus-secreting goblet/secretory cells and ciliated cells, while facilitating the formation of tight junctions and other cell–cell interactions that are essential for polarization, thereby enhancing the epithelial barrier’s integrity [132,133]. For instance, McQueen et al. (2020) successfully utilized a membrane insert-based human fallopian tube model with ALI to induce epithelial polarization towards mucin-secreting goblet cells and multi-ciliated epithelial cells [134]. This model enabled the investigation of Ct inclusion formation, as well as the dynamics of chlamydial growth and replication within host cells.

#### 3.2.3. Co-Culture

To address the limitations of conventional mono-cell culture, a variation of the cell culture technique has been developed, involving co-culture systems. They incorporate two or more distinct cell populations that interact either directly or indirectly with one another [135,136,137]. While co-cultures can be grown to form a monolayer, they present challenges, such as optimizing the ratio of different cell types and selecting an appropriate culture medium that can support both cell types. Cell culture membrane inserts within the context of co-cultures can alleviate some of these complications [135].

These systems are widely applied in both viral and bacterial infectious disease studies, offering a more accurate representation of host–pathogen interactions. For example, Barreto-Duran et al. utilized a membrane insert-based co-culture consisting of the primary human airway epithelium and human monocyte-derived macrophages to investigate SARS-CoV-2 infection. This model revealed the importance of neutrophils in enhancing pro-inflammatory cytokine release, impairing epithelial barrier integrity, and increasing the infection of basal stem cells during the infection response [138]. The principle of this approach is equally relevant to Ct research. Given that Ct infection involves intricate interactions between various epithelial and immune cell types and signaling pathways, implementing co-culture models in Ct studies can provide critical insights into pathogen persistence and immune evasion [115].

More recently, there have been a growing number of studies on Ct infections that incorporate an underlying stromal component within membrane insert-based systems. Stromal cells play a crucial role in regulating epithelial function through epithelial–stromal interactions by maintaining the tissue architecture, supporting the epithelial morphology, modulating the immune responses, and facilitating oxygen and nutrient delivery [139]. These interactions are particularly significant in the context of Ct infections, where the stroma can influence the epithelial response to infection. For instance, Hall et al. (2011) developed a co-culture system where Ishikawa cells or HEC-1B endometrial epithelial cells were cultured apically, while SHT-290 stromal cells were placed on the bottom of the well plate [140]. This setup revealed that soluble factors released by estrogen-stimulated stroma cells enhanced the susceptibility of the epithelial cells to Ct serovar E. Nogueira et al. (2017) developed a 3D stratified squamous epithelial model by culturing HaCaT keratinocyte epithelial cells on fibroblasts embedding collagen gel under an ALI [81]. This organotypic epithelium allowed both terminally differentiated and undifferentiated cells, exhibiting varying susceptibility to Ct serovars L2 and D. Similarly, Edwards et al. (2019) developed a cervicovaginal model with fibroblasts, with rat tail collagen added to the basal surface of the membrane, providing an extracellular matrix (ECM) environment, while A2EN servicovaginal epithelial cells were cultured in the apical compartment [141]. Both the apical and basolateral compartments were exposed to vaginal *Lactobacillus* spp., before inoculation with Ct. This model highlighted the importance of host–microbiota interactions in modulating the protective mechanisms against CT infection. Building on this concept, subsequent research introduced a vaginal model, providing new insights not only into Ct but also into *Neisseria gonorrhoeae* infections [142].

While the membrane insert system supports the co-culture of multiple cell types, it still faces many of the key limitations associated with monolayer cultures, such as the restriction to a 2D topology that does not reflect the 3D anatomy and the impact of rigid polymer substrates on cell behavior. The higher cost of the membrane inserts compared to standard tissue culture plates is another significant barrier to its widespread use. In addition, the pores in the membrane can lead to unintended exosome transfer between two compartments, which may confound the resulting analysis [143].

### 3.3. Three-Dimensional Models and Organoids

Over the past decade, a diverse array of three-dimensional (3D) models have emerged to address the limitations inherent in traditional monolayer systems. These advanced models aim to more accurately recapitulate the complex cellular interactions and spatial organization of the female reproductive tract, thereby providing more physiologically relevant platforms for the study of Ct’s infection dynamics.

A crucial factor that has enabled the recent uptick in 3D model advancement has been the development, refinement, and commercial availability of hydrogels. These hydrophilic polymers are crosslinked to form a 3D network, providing a crucial microenvironment that can guide cell behavior. They contain high water content, resembling the hydrated native ECM [144]. While hydrogel’s fibrillar architecture provides structural support, their porosity facilitates the diffusion of essential gases and nutrients that sustain cell survival and growth [145,146]. Moreover, the diversity of natural, synthetic, and composite hydrogels offers flexibility to impose varying degrees of structural and functional modification tailored to specific research questions [147]. This has enabled more robust cellular expansion and directed differentiation, which was previously unattainable with conventional two-dimensional (2D) culture systems.

A prime example of a biomimetic 3D model is represented by organoids, which are self-assembling 3D cellular constructs that closely recapitulate the architecture and functionality of the source tissue and are often cultured within mouse tumor-derived hydrogels [148]. A key advantage of organoids is their ability to preserve the natural cellular diversity and heterogeneity of native tissues, maintaining both stem and differentiated cell populations [149]. Moreover, organoids can be continually expanded for extended periods without losing their phenotype—potentially indefinitely under optimal conditions. This enables longer experimental timeframes that are not feasible with standard cell culture models [150].

The improved physiological and stable long-term maintenance of organoids make them an invaluable tool for the study of infectious diseases, including Ct infections. Unlike 2D monolayers, organoids preserve critical features such as cellular polarity, diversity, and the 3D architecture, which can facilitate the more accurate assessment of host–pathogen interactions over time [151,152,153]. These characteristics enable studies on chronic Ct infections and provide insights into epithelial barrier function and pathogen invasion strategies. For instance, Kessler et al. (2019) demonstrated the innovative use of human fallopian tube organoids in examining the long-term effects of Ct serovars D, K, and E on the epithelial cell phenotype and integrity [154]. In this study, Ct infection was introduced by breaking up the organoids, pelleting the cells with Ct, and allowing the regrowth of the organoids.

This organoid-based approach has been extended to other female reproductive systems, such as murine oviduct organoids, murine endometrial organoids, and human endometrial and cervix organoids [155,156,157,158,159]. Notably, Dolat et al. (2022) introduced Ct directly into the endometrial organoids by microinjection, revealing that Ct exploits tight junction breakdown to disrupt the epithelial barrier, potentially gaining access to basolateral receptors via the bacterial protein TepP [158]. These findings highlight the advantage of the spheroid 3D structure of organoids in visualizing pathogen–host interactions.

Despite these advantages, organoids have limitations that affect their physiological relevance. While it can present visual cues for epithelial function, the spheroid form does not fully mimic the structure of the tissue. In turn, the unique infection pattern within the female reproductive tract cannot be emulated. For example, ascending Ct typically enters the fallopian tube from one end and interacts with the epithelium along its length, a process that is poorly replicated in organoid models [153,160]. The infection techniques used, such as microinjection or pelleting organoid fragments with Ct, often result in inconsistent microbial-to-cell ratios and uneven infection dynamics across organoids [153,161]. Finally, as described in other in vitro models above, organoids lack systemic interactions, such as an underlying stromal layer or an integrated and functional vascular network [162]. Consequently, while organoids provide a solid foundation for infection modeling, their limitations emphasize the need for increasingly sophisticated models that better simulate tissue structures and systemic responses.

## 4. Discussion

Chlamydia infections pose significant challenges to women’s health and reproductive outcomes. Research efforts to elucidate the pathogenesis of chlamydial infections have grown significantly over the past few decades, utilizing diverse in vivo and in vitro models.

As discussed in this review, animal models provide valuable insights but face significant challenges, including species-specific differences, high costs, ethical concerns, and limited translational relevance. Similarly, conventional 2D in vitro systems have been instrumental in understanding host–pathogen interactions but lack the dynamic complexity of the female reproductive system. Recent advancements in in vitro modeling, such as incorporating multiple cell types, increasing the dimensionality, and adopting biocompatible hydrogel materials, have improved our ability to mimic the human tissue architecture and functions. These innovations facilitate detailed investigations into the infection dynamics, immune responses, and epithelial barrier disruptions. However, replicating the breadth of cues observed in natural tissues remains a challenge.

Emerging technologies on the horizon, particularly organ-on-chip (OoC) platforms, address some of these limitations by recreating dynamic multicellular microenvironments. Studies on various OoC systems have emphasized the importance of incorporating biomechanical factors such as fluid flow, diffusion gradients, and shear stress in maintaining the epithelial integrity and regulating host–pathogen interactions, thereby enabling models to closely mimic the human physiology [163,164,165,166,167]. In particular, reproductive-tract-on-chip models, including endometrium- and oviduct-on-chip systems, have demonstrated their potential in the study of Ct–host interactions under physiologically relevant conditions [168,169,170]. However, technical challenges like miniaturization constraints and limited analytical tools must be overcome to fully realize the potential of OoC technology.

In silico approaches, including computational modeling and artificial intelligence (AI), represent a complementary frontier in Ct research. Computational models can simulate the infection dynamics, predict vaccine efficacy, and optimize the experimental parameters, while AI facilitates the analysis of large datasets, including genetic and imaging data [166,171,172,173,174,175,176]. Integrating OoC platforms with in silico methods can enable scalable, cost-effective, and accurate models for the study of Ct’s pathogenesis.

As shown, advances in interdisciplinary technologies will deepen our understanding of the chlamydial pathophysiology and enable the development of standardized and reproducible Ct study models tailored to specific research objectives. Ultimately, we believe that the synergy between diverse research methodologies from interdisciplinary fields will enable us to enhance the efficiency, accuracy, and practicality of complex Ct models, particularly in resource-limited settings.

## Figures and Tables

**Figure 1 microorganisms-13-00553-f001:**
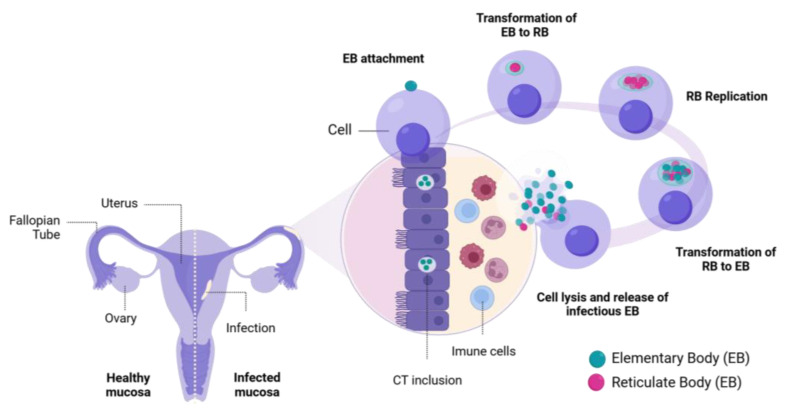
The developmental cycle of *Chlamydia trachomatis* in the female reproductive tract. Created with BioRender.com accessed on 23 December 2024.

**Figure 3 microorganisms-13-00553-f003:**
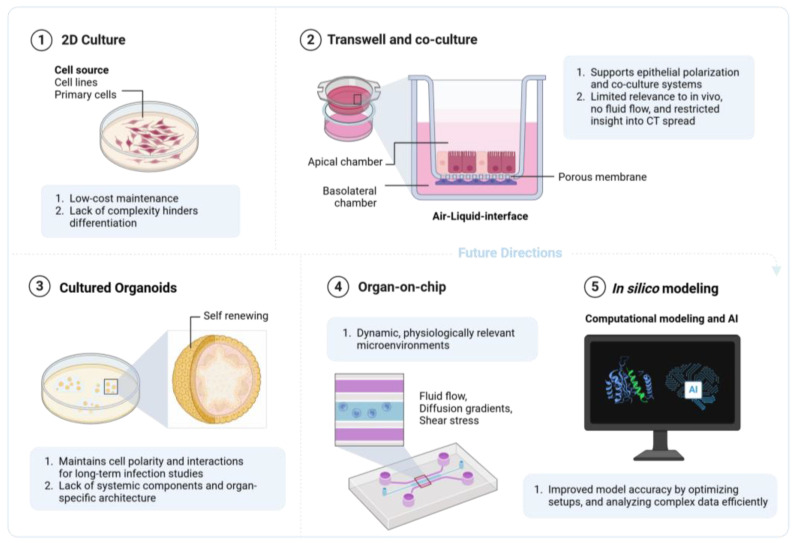
Evolution of in vitro models for the study of *Chlamydia trachomatis* infections in the female reproductive tract. Created with BioRender.com accessed on 23 December 2024.

**Table 1 microorganisms-13-00553-t001:** Summary of animal models used to study *Chlamydia trachomatis* infections of the genital tract, with their main advantages and limitations.

Model	Advantages	Limitations
Mouse	Well-characterized inbred and knockout mouse strains available Low cost and easy handling Limited space needed	Genetic homogeneity among laboratory strains, not suitable to study patient-to-patient variability Difficulty in studying long-term consequences of Ct infections and infections of upper genital tract Results not always directly translatable to human diseases
Pig	Comparable size, similar physiology and immune system to humans High genetic similarity to humans Ethically more acceptable than non-human primates	Some differences in genital tract compared to humans Knockout or transgenic pigs not widely available Limited number of animal-specific immunological reagents Higher costs associated with animal maintenance and required space compared to rodents
Guinea pig	Higher similarity to female genital tract of human, compared to mouse models Ct infections can be transmitted through sexual contact Low cost and easy handling Limited space needed	Smaller size and some differences in genital anatomy and estrous cycles compared to humans Differences in vaginal microbiota composition compared to humans Limited number of animal-specific immunological reagents
Non-humanprimate	Close anatomical and physiological resemblance to humans High genetic similarity to humans Menstrual cycle and genital microbiota similar to humans Susceptibility to urogenital Ct infections, including chronic infections	High costs associated with animal maintenance and required space Ethical considerations Strict regulations in Europe for research involving non-human primates

## Data Availability

Not applicable.

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
