# Peer review of "Study Models for Chlamydia trachomatis Infection of the Female Reproductive Tract"

_microorganisms, 2025, doi:10.3390/microorganisms13030553_

Round 1
Reviewer 1 Report
Comments and Suggestions for Authors
Dear Authors,
I read the manuscript "Study Models for Chlamydia trachomatis infection of the female reproductive tract
Jaehyeon Kim et al. In general, the paper is interesting even though the literature contains many similar reviews. However, the manuscript should be improved in some details for a possible publication.
Criticism
1. to be more detailed, the Chlamydia Spp cycle should also include the ‘aberrant corpuscles’ that came from the reticulated bodies that do not transform into the elementary bodies; this condition explains Chlamydial persistence and the development of chronic clinical chronic forms. Please read and cite Beatty WL, Morrison RP, Byrne GI (1994) Persistent Chlamydiae: from cell culture to a paradigm for Chlamydial pathogenesis. Microbiol Rev 58(4):686–699; Behrens-Baumann W (2007) Chlamydial diseases of the eye. A short overview. Ophthalmologe
104(1):28–34.
2. Authors state that "A key limitation is that Ct does not naturally infect mice, making it difficult to fully
replicate the progression of disease seen in humans". Please, add one or two references.
3.line 154-155. 2.2. Porcine (Pig) models: this paragraph should be shortened (30%). Also, (line 183 .....) shorten the 2.3. Guinea pig models paragraph (30%).
4. A table should be built to highlight the differences of Chlamydia infection in the porcine models as well as in mouse models and non-human primate models.
5. Figure 1 and 2 are fine
6. The paragraphs on cultivation aspects (3.1 and 3.2) should be shortened (20 %).
7. Lines 402-405: this concept should also be deepened with reference to humans and moved into the discussion
Author Response
Comment 1: to be more detailed, the Chlamydia Spp cycle should also include the ‘aberrant corpuscles’ that came from the reticulated bodies that do not transform into the elementary bodies; this condition explains Chlamydial persistence and the development of chronic clinical chronic forms. Please read and cite Beatty WL, Morrison RP, Byrne GI (1994) Persistent Chlamydiae: from cell culture to a paradigm for Chlamydial pathogenesis. Microbiol Rev 58(4):686–699; Behrens-Baumann W (2007) Chlamydial diseases of the eye. A short overview. Ophthalmologe 104(1):28–34.
Response 1: We thank the author for noticing this oversight in our description of the Ct cycle. We have amended the text to include the additional suggested details and citations
Comment 2: Authors state that "A key limitation is that Ct does not naturally infect mice, making it difficult to fully replicate the progression of disease seen in humans". Please, add one or two references.
Response 2: Supporting references have been affixed to this statement. These citations were cited further down in the text, but we have now moved the citations to clarify exactly from which sourcse the statement has been derived. Specifically, citation 34 through 37:
(34) Lyons, J. M.; Ito, J. I.; Morré, S. A. Chlamydia Trachomatis Serovar E Isolates from Patients with Different Clinical Manifestations Have Similar Courses of Infection in a Murine Model: Host Factors as Major Determinants of C Trachomatis Mediated Pathogenesis. J Clin Pathol 2004, 57 (6), 657–659. https://doi.org/10.1136/jcp.2003.013086.
(35) Gondek, D. C.; Olive, A. J.; Stary, G.; Starnbach, M. N. CD4+ T Cells Are Necessary and Sufficient To Confer Protection against Chlamydia Trachomatis Infection in the Murine Upper Genital Tract. The Journal of Immunology 2012, 189 (5), 2441–2449. https://doi.org/10.4049/jimmunol.1103032.
(36) Sturdevant, G. L.; Caldwell, H. D. Innate Immunity Is Sufficient for the Clearance of Chlamydia Trachomatisfrom the Female Mouse Genital Tract. Pathogens Disease 2014, 72 (1), 70–73. https://doi.org/10.1111/2049-632X.12164.
Comment 3: line 154-155. 2.2. Porcine (Pig) models: this paragraph should be shortened (30%). Also, (line 183 .....) shorten the 2.3. Guinea pig models paragraph (30%).
Response 3: We agree with the reviewer that some of the included information in 2.2 and 2.3 could be streamlined. As such, we have amended the text to be more concise, while still retaining the important information relevant to these sections (please see the Track Changes version of the resubmitted manuscript for a complete overview).
Commet 4: A table should be built to highlight the differences of Chlamydia infection in the porcine models as well as in mouse models and non-human primate models.
Response 4: We appreciate the suggestion of the reviewer and share their perspective that a summarized table is a useful addition for readers to quickly assess key comparative points. We have therefore added “Table 1.Summary of animal models”
Comment 5: Figure 1 and 2 are fine
Response 5: We appreciate the reviewers comment regarding our use of graphics and recognition of our time spent in their preparation.
Comment 6: The paragraphs on cultivation aspects (3.1 and 3.2) should be shortened (20 %).
Response 6: Again, we agree with the reviewer that it is important to provide readers with a concise overview. We have systematically streamlined the text while keeping the core message of these sections. (please see the Track Changes version of the resubmitted manuscript for a complete overview).
Comment 7: Lines 402-405: this concept should also be deepened with reference to humans and moved into the discussion
Response 7: We have expanded on this topic and added suitable references. We discuss more generally the role of a multicellular environment on infections, citing other examples from Covid research and then further discuss this within the specific context of Ct infections. While we understand this point is particularly interesting, we felt that this expanded discussion was still most appropriately located within Section 3.2 because of its relevance to the development of more complex in vitro models. We have exclusively focused on future directions for research models within our Discussion section, in line with the overall theme of our manuscript, and thus felt that including this particular point within that section would not be aligned with the manuscript structure.
Reviewer 2 Report
Comments and Suggestions for Authors
Overall, it is a good idea to write a manuscript describing models for studying female reproductive tract infections caused by Chlamydia trachomatis.
However, the reviewer has several questions and comments regarding the manuscript.
Lines 155-157. Here the authors point out significant similarities in the genital anatomy, hormonal cycles, and immune systems of humans and pigs, which allows the pig to be used as a model for studying chlamydia.
There is a contradiction here; the authors themselves, just below (162-165), write that humans and pigs have anatomical, histological, and cytological differences in the structure of the lower reproductive tract. Moreover, there are differences in the hormonal cycle; the reviewer wanted to remind us that pigs have an estrous cycle, while humans have a menstrual cycle. These are different phenomena; apparently the authors should not talk about the similarity of hormonal cycles here. The reviewer suggests that the authors reformulate this section of the text.
Section 3 (In vitro models). Here, considerable attention is paid to in vitro models. A shortcoming of this section is the use of literature outside the context of chlamydia and chlamydias infection; the authors cite works that are completely unrelated to the problem being described; for example, [81] describes the role of hormones in susceptibility to HIV, [83] - the role of the microbiome in susceptibility to infections, [86] is devoted to the intestinal epithelium outside the context of chlamydia, as are [118] and [166]. In general, this section is distinguished by a detailed description of many features of cell cultures; nevertheless, the reviewer would like to note that the data described are not always relevant to the stated purpose of the authors in writing this manuscript. Obviously, here it would be necessary to dwell in more detail on the analysis of publications where cell cultures were used as models specifically for studying chlamydias infection. In connection with the above, the reviewer suggests adjusting the description of this section so that its content corresponds to the general spirit of the manuscript.
Author Response
Comment 1: Lines 155-157. Here the authors point out significant similarities in the genital anatomy, hormonal cycles, and immune systems of humans and pigs, which allows the pig to be used as a model for studying chlamydia.
There is a contradiction here; the authors themselves, just below (162-165), write that humans and pigs have anatomical, histological, and cytological differences in the structure of the lower reproductive tract. Moreover, there are differences in the hormonal cycle; the reviewer wanted to remind us that pigs have an estrous cycle, while humans have a menstrual cycle. These are different phenomena; apparently the authors should not talk about the similarity of hormonal cycles here. The reviewer suggests that the authors reformulate this section of the text.
Response 1: We thank the reviewer for pointing out this contradiction. We have revised the text to be more specific regarding the general similarities (e.g.g the size and gross anatomy) compared to the minutiae which distinguish the porcine and human reproductive tracts (e.g. cellular composition and tissue microstructure). We have also removed the references to similarities in hormone cycles (please refer to the track change in the resubmission package for complete details).
Comment 2: Section 3 (In vitro models). Here, considerable attention is paid to in vitro models. A shortcoming of this section is the use of literature outside the context of chlamydia and chlamydias infection; the authors cite works that are completely unrelated to the problem being described; for example, [81] describes the role of hormones in susceptibility to HIV, [83] - the role of the microbiome in susceptibility to infections, [86] is devoted to the intestinal epithelium outside the context of chlamydia, as are [118] and [166]. In general, this section is distinguished by a detailed description of many features of cell cultures; nevertheless, the reviewer would like to note that the data described are not always relevant to the stated purpose of the authors in writing this manuscript. Obviously, here it would be necessary to dwell in more detail on the analysis of publications where cell cultures were used as models specifically for studying chlamydias infection. In connection with the above, the reviewer suggests adjusting the description of this section so that its content corresponds to the general spirit of the manuscript.
Response 2: We thank the reviewer for our oversight in clarifying many of the references we have used in this section and for providing a detailed list of points for us to address. In general, our use of citations that stem from seemingly unrelated fields is intentional, since our focus in these moments is not in the application of the in vitro models to Ct research but to discuss the current state of the art in the field of in vitro models. Unfortunately, many of the latest innovations in 3D in vitro models which would be relevant for studying Ct have yet to be applied to this particular field. Meanwhile, many promising examples come from the field of intestinal models and lung models. By including these references, we hope to inspire the field of Ct research to embrace these developments from other fields.
However, we acknowledge that their inclusion can be misleading. In that regard, we have done our utmost to replace any references that appear to be off topic with a Ct-related publication. Specifically, [116] (was [118]) has been replace with a more relevant citation. And [118] has been removed entirely. Where this was not possible, because of a lack of existence of such models, we explicitly mention to which tissues and diseases the described in vitro developments are associated with and the relevance this has for Ct research. There are also a few reviews we have cited (for example, [87] (was [86])) which are not specifically on Ct but provide an excellent review of general topics related to in vitro models; again, while these might not be Ct-oriented, we feel their use in this context is still relevant and helps readers better understand what might be possible with in vitro systems.
Reviewer 3 Report
Comments and Suggestions for Authors
The authors present an overview of recent trends in Ct research, detailing the various in vivo and in vitro models to study the pathogenesis of chlamydial diseases. They detail the pros and cons of these models. The authors also discuss how many of these models will be valuable in the future for more in-depth research. I consider that the manuscript is thorough, of significant scientific value, and will be of interest to readers.
Author Response
Comments 1: The authors present an overview of recent trends in Ct research, detailing the various in vivo and in vitro models to study the pathogenesis of chlamydial diseases. They detail the pros and cons of these models. The authors also discuss how many of these models will be valuable in the future for more in-depth research. I consider that the manuscript is thorough, of significant scientific value, and will be of interest to readers.
Response 1: We thank the reviewer for their kind words and positive assessment of our manuscript.
Round 2
Reviewer 1 Report
Comments and Suggestions for Authors
Dear Editor,
I had already seen and evaluated the manuscript “Study Models for Chlamydia trachomatis infection of the female reproductive tract by Kim J et al.
It does not appear to me that the changes I had asked for regarding the length of the paragraphs2.1, 2.3, 2.4. have been made. Table required is here and it's fine
Dear Editor,
I had already seen and evaluated the manuscript “Study Models for Chlamydia trachomatis infection of the female reproductive tract by Kim J et al.
It does not appear to me that the changes I had asked for regarding the length of the paragraphs have been made.
Suitable for publication.
Kind regards
Author Response
Comments 1: It does not appear to me that the changes I had asked for regarding the length of the paragraphs2.1, 2.3, 2.4. have been made.
Response 1: We thank the reviewer for their follow up assessment. To clarify, the previous comment from the reviewer suggested to shorten the porcine and guinea pig sections.
"Porcine (Pig) models: this paragraph should be shortened (30%). Also, (line 183 .....) shorten the 2.3. Guinea pig models paragraph (30%)."
We would like to point out that our first revision permitted us to streamline both of these sections. The Porcine section was originally 339 words, which we revised to 275 words (approximately 20% shorter). Meanwhile, the Guinea pig section was reduced from 346 to 304, which is admittedly only a reduction of 10%. At the moment, murine models have about 570 words, reflecting the large number of mouse studies in the field, while the porcine, guinea pig, and non-human primates have 275, 304, and 320, respectively. We feel these proportions maintain balance between the different models in terms of their relative quantity and impact while still providing enough information for effective comparison.
While we very much appreciate the emphasis on brevity and were grateful for the opportunity to make these sections more concise, we feel that further reduction would compromise the key points we are aiming to make and as well negatively affect the readability of these sections. Therefore, we would respectfully ask to maintain these sections in their currently revised state.